# Tumor Chemosensitivity Assays Are Helpful for Personalized Cytotoxic Treatments in Cancer Patients

**DOI:** 10.3390/medicina57060636

**Published:** 2021-06-19

**Authors:** Engin Ulukaya, Didem Karakas, Konstantinos Dimas

**Affiliations:** 1Department of Clinical Biochemistry, Faculty of Medicine, Istinye University, Istanbul 34010, Turkey; 2Department of Molecular Biology and Genetics, Faculty of Science and Letters, Istinye University, Istanbul 34010, Turkey; didem.karakas@istinye.edu.tr; 3Department of Pharmacology, Faculty of Medicine, University of Thessaly, 41500 Larissa, Greece; kdimas@med.uth.gr

**Keywords:** tumor chemosensitivity assay, drug response assay, treatment, pharmacogenetics

## Abstract

Tumor chemosensitivity assays (TCAs), also known as drug response assays or individualized tumor response tests, have been gaining attention over the past few decades. Although there have been strong positive correlations between the results of these assays and clinical outcomes, they are still not considered routine tests in the care of cancer patients. The correlations between the assays’ results (drug sensitivity or resistance) and the clinical evaluations (e.g., response to treatment, progression-free survival) are highly promising. However, there is still a need to design randomized controlled prospective studies to secure the place of these assays in routine use. One of the best ideas to increase the value of these assays could be the combination of the assay results with the omics technologies (e.g., pharmacogenetics that gives an idea of the possible side effects of the drugs). In the near future, the importance of personalized chemotherapy is expected to dictate the use of these omics technologies. The omics relies on the macromolecules (Deoxyribonucleic acid -DNA-, ribonucleic acid -RNA-) and proteins (meaning the structure) while TCAs operate on living cell populations (meaning the function). Therefore, wise combinations of TCAs and omics could be a highly promising novel landscape in the modern care of cancer patients.

## 1. Introduction

Chemotherapy is still an important option for the treatment of various cancer types. It is especially unavoidable when the disease evolves into the metastatic stage. However, there is still no satisfactory improvement in the clinical outcome of cancer patients, although many chemotherapeutic drugs have been introduced in the last 50 years. Since cancer is a heterogeneous disease, the responses to the same chemotherapeutic may vary from one patient to another one, even if cancer cells are originated from the same organ (the histologically same phenotype). To overcome this difficulty, the treatments should be personalized (tailored chemotherapy). For this purpose, genome-based methodologies have increasingly been developed and seem to be highly appreciated by the oncology community, but it is still not sufficient to effectively treat the patients [1]. Some of these approaches (e.g., liquid biopsy for mutational status) are not cost-effective, however. There are numerous other kinds of approaches that keep being introduced to the oncology field. These approaches rely on functional tumor biology and are known as tumor chemosensitivity assays (TCAs). Considering the importance of the knowledge that will be produced from genome-based technologies (e.g., pharmacogenetics) and functional tumor biology-based assays, it could be wise idea to combine these assays (Figure 1). Either type of these assays is not routinely used in the clinics for various reasons, such as a lack of randomized controlled prospective clinical trials.

Each assay type has pros and cons. TCAs have a satisfactory negative predictive value (showing the possibility of drug resistance), ranging from roughly 66% to 95%, and a moderate positive predictive value (showing the possibility of drug sensitivity), ranging from roughly 50% to 85% [2,3]. Pharmacogenetics is, to some extent, able to give some clues for the possibility of the side effects of the drugs although it is still not in clinical use. However, the combination of TCAs and pharmacogenetics could provide the oncologists with useful data to manage the disease. For example, while TCAs could allow the elimination of ineffective drugs (due to the high negative predictive value), pharmacogenetics could help the oncologists to choose the best possible drug options amongst the tested drugs by the TCA test.

The objective of this review is to summarize the role and importance of TCAs in personalized medicine by discussing the literature. Besides, different techniques to evaluate TCAs have been reviewed considering their pros and cons.

## 2. Tumor Chemosensitivity Assays (TCAs)

TCAs have been designed to select the most appropriate chemotherapy option for individual cancer patients by indicating resistance or sensitivity for drugs. Over the years, TCAs have made satisfactory progress, evolving from some technologically simple assays (e.g., clonogenic assay) to technologically advanced assays (e.g., luminescence-based assays like ATP-TCA or organoids). Thanks to technological achievements, even microfluidic systems have been started to be used for TCAs. As an example, the results of a study performed with a microfluidic system that has been used for real-time screening of drugs in a 3D environment indicated that the microfluidic system detects the resistance and sensitivity of cancer cells to chemotherapeutics in less than 12 h [4]. Moreover, a patient-specific mathematical model has been created to be used to perform TCA and allows one to predict the clinical response to up to 31 drugs within 5 days after bone marrow biopsy [5].

TCAs are able to guide oncologists about which anti-cancer drugs are more likely to work well enough (or not) on a given patient [6]. Some reports have shown that TCAs could help oncologists to predict the response rate, recurrence, platinum-based drug resistance, the progression-free survival as well as the overall survival rate of cancer patients [7,8,9,10,11,12]. TCAs have been successfully used to select the better treatment regimen in patients with colorectal liver metastasis and it was shown that the assay-based treatment led to better response rates to drugs [13]. Some reports in the literature clearly show that there is a correlation between the results of TCAs and patient outcomes [14,15,16]. Moreover, it was reported that there is a good correlation between the genes predicted to be involved in mechanisms of drug sensitivity/resistance and in vitro chemosensitivity [17]. This is of particular importance in the definition of predictive signatures to guide individualized chemotherapy. Taking the above into account, TCAs keep gaining attention as is proven by the increasing number of TCA-based publications over the years.

There are various ways for performing TCAs for the prediction of response to treatment. The basic idea is to take the tumor tissue from the patient and isolate the tumor cells at the laboratory and then choose one of the TCAs of interest. To perform the test, the cancer tissue removed from patients during a routine surgical operation or taken by a biopsy should be sent to the laboratory in a special transport medium that keeps the cells alive. After that, as many as up to 20, even 30, different drugs with different doses are incubated with the cancer cells isolated from the fresh tissue in order to determine their cytotoxic activity (Figure 2).

Since the 1950s, several methods have been developed to evaluate tumor chemosensitivity/chemoresistance. Some well-known assays are listed in Table 1. The first three methods are given in detail in the next sections.

### 2.1. The Human Tumor Clonogenic Assay

Clonal growth of mammalian cells was first achieved in 1950s [18] and based on the clonal growth capability of tumor cells, the human tumor clonogenic assay (HTCA) was firstly tested by Hamburger and Salmon in 1977 [19]. HTCA is a soft agar system designed for growing tumor tissues in cell culture conditions and thus allows one to have a prediction about the responses to chemotherapy in cancer patients [20].

In the 1980s, HTCA was used to determine drug response in different types of cancer, such as uroepithelial cancers including prostate, testicular, renal [21], and ovarian, breast and colorectal cancer [22]. Based on the results of these studies, the authors suggested that the HTCA might be a useful tool for the evaluation of anti-tumor effects of drugs in vitro. However, as is mentioned above, this method was used to determine drug response (sensitive/resistance), especially in the 1980s, and it seems to be losing attention. The main reason for this is that it is time-consuming, subjective and not suitable for automation. For the reasons above, this type of assay seems not to be helpful anymore in terms of the prediction of response to the chemotherapy in patients, but it has still had a great value in stem cell-based research activities.

### 2.2. MTT (3-(4,5-Dimethylthiazol-2-yl)-2,5-Diphenyltetrazolium Bromide) Based Chemosensitivity Assays

The MTT assay is a colorimetric cell viability assay for assessing the metabolic activity of cells. The principle of the assay is reduction of MTT tetrazolium salt to a blue/purple formazan crystal by living cells but not by dead cells. Then, the concentration of dissolved formazan crystals can be quantified using a spectrophotometer and it is in direct correlation to the number of metabolically active cells [23,24,25]. The MTT assay represents a simple and rapid colorimetric assay and yields quantitative data [26].

The MTT assay can be used to measure chemosensitivity of tumor cells [26,27,28]. MTT-based chemosensitivity assays were evaluated in several types of cancer such as childhood leukemia [28], brain [29], colorectal [30], acute myeloid leukemia [16], lung [31], and ovarian cancer [32].

However, the MTT assay has some disadvantages. For example, damaged or inactivated mitochondria are able to produce formazan crystals [33,34,35], some chemicals can interact with MTT salt resulting in false results in viability [36,37], or tested agents may interfere with mitochondrial dehydrogenase activity, resulting in activation or inhibition of mitochondrial dehydrogenases and thus over/underestimation of the MTT assay results [38]. Therefore, any tumor chemosensitivity/chemoresistance data in the literature obtained from the MTT assay should be interpreted with great caution to avoid false-positive/negative results.

### 2.3. ATP-Based Chemosensitivity Assay (ATP-TCA)

The ATP assay was first developed by Lundin and colleagues as a somatic cell viability assay [39]. The ATP-TCA is a standardized system which can be adapted to a variety of uses with both cell lines and primary cell cultures. Generally, tumor chemosensitivity/chemoresistance assays use the culture of tumor cells in vitro, and this method may also cause stromal cell contamination in the tested sample. The response of stromal/epithelial cells to chemotherapy may greatly differ and thus contamination of stromal cells may cause to the unsuccessful treatment because of misinterpreted results. Therefore, serum-free culture medium and polypropylene plates are used in this method to prevent the growth of non-neoplastic cells over a 6-day incubation period. Then, intracellular ATP is extracted by a detergent-based lysis solution and relative ATP levels are measured as bioluminescence light by a luciferin–luciferase reaction [40]. The reaction is briefly summarized in Figure 3.

Sevin and his coworkers were the first to adapt this method to be used for solid human tumor specimens [41], then numerous works were performed by this assay. ATP-based chemosensitivity assays have been evaluated on many tumor types such as pancreatic cancer [42], ovarian cancer [43], colorectal [44], breast [45], non-small cell lung cancer [46]. Many studies have shown the reliability of ATP as a sensitive measure of cell viability for various cell lines.

ATP-TCA seems to be useful particularly in ovarian cancer. In a study performed with ovarian carcinoma cells, it was reported that ATP-TCA measured cisplatin resistance with >90% accuracy [47]. The authors suggested that the ATP-based chemosensitivity/chemoresistance assay has a high sensitivity, linearity, and precision for measuring the activity of single agents and drug combinations [47].

In another study, a significant correlation was reported between drug sensitivity/resistance and clinical response (*p* = 0.007) [15]. In the same study, the assay demonstrated a sensitivity, specificity, positive predictive value (PPV), and negative predictive value (NPV) of 95%, 44%, 66%, and 89%, respectively. The NPV in this study is especially notable because it gives an appreciable idea of the patients who will show recurrence (due to drug resistance) within 12 months of the post-chemotherapy period. In a recent study, the sensitivity, specificity, positive predictive value and negative predictive value for the clinical chemotherapy sensitivity were found to be as 88.6%, 77.8%, 83% and 84.8% in ovarian cancer patients, respectively [2]. These values should be considered acceptable and promising for the future of ATP-TCA. The other study conducted on patients with unresectable non-small cell lung cancer showed that there was a good correlation between ATP-TCA results and clinical outcomes of patients [46]. Another study analyzed the correlations between the clinical outcomes of patients and tumor chemosensitivity assay in stage III lung cancer patients. The results indicated that the disease-free survival rates were detected longer in patients with higher drug sensitivity compared to patients who display drug resistance (18 vs. 8.5 months, *p* < 0.05) [48].

ATP-TCA even reached a randomized clinical study [7]. In this study, the results of a small randomized clinical trial had documented a trend towards improved response and progression-free survival for assay-directed treatment chemosensitivity testing [7]. The same research group also published other studies in later years in which they stated that the genes predicted to be involved in drug sensitivity and resistance correlate well with in vitro chemosensitivity in ovarian cancer [49] and may allow the definition of predictive signatures to guide individualized chemotherapy in lung cancer [17].

ATP-TCA has been further studied in a more advanced fashion in a deadly skin disease, malignant melanoma. In a study performed by Ugurel et al., 2006, they have found that the objective response was 36.4% in chemosensitive cases in comparison with 16.1% in chemoresistant cases (*p* = 0.114); a good outcome was seen in 59.1% versus 22.6% (*p* = 0.01) [50]. Furthermore, the chemosensitive cases had an increased overall survival of 14.6 months, compared with 7.4 months in chemoresistant cases (*p* = 0.041). The same group recently published another study in malignant melanoma [51]. In their study, they compared the sensitivity index values of dacarbazine-treated patients with classical chemotherapy (cisplatin, paclitaxel, treosulfan, gemcitabine) agent-treated patients. They found that there was no significant difference between these two different therapy protocols in terms of the superiority of any protocol over the other. However, classical chemotherapy-given patients developed more severe side effects compared to the dacarbazine protocol. In addition, ATP-TCA may help to reduce the number of phase II/III trials, saving money and time, while permitting new agents effective for subsets of cancer patients to be introduced faster. In conclusion, ATP-TCA seems to provide promising retrospective ex vivo/in vivo correlations with acceptable positive and negative predictive values. Also, ATP-TCA provides a rational approach for preclinical evaluation of novel active combination regimens for diverse malignant diseases. More importantly, ATP-TCA-directed chemotherapy can produce impressive response rates although the long-term results (progression-free and overall survival) may suffer from the lack of satisfactory results. Also, ATP-TCA-directed chemotherapy appears to be of particular value for platinum-refractory patients. However, more clinical studies are still needed for firm conclusions, although the correlations between the assay results and response to treatment are highly promising. ATP-TCA is a well-accepted and promising methodology as it provides reliable data in hematological cancers as well. In a study, it has recently been reported that ATP-TCA demonstrated a significant correlation with complete response for chemotherapy and could be a useful tool to optimize personalized treatments for patients with acute myeloid leukemias [52]. ATP-TCA is able to predict not only the best chemotherapy regimen to any given patient but also the survival rate. Accordingly, in a recent study, it was shown that patients with higher drug sensitivity tended to have longer disease-free survival (18 vs. 8.5 months) than patients displaying drug resistance [48]. Taken altogether, ATP-TCA might be considered as a helpful methodology in terms of obtaining better therapy outcomes in cancer patients.

In addition to methods which we mentioned above, some other techniques are also used to test drug resistance and/or sensitivity such as extreme drug resistance assay [53,54,55], tissue explant assay [56,57], differential staining cytotoxicity assay [25,58,59], fluorescent cytoprint assay [60,61], collagen-gel droplet embedded culture drug sensitivity test [62,63,64] (Table 1).

## 3. Role of Cancer Stem Cells in Tumor Chemosensitivity Assays

Cancer patients with the same stage and grade may respond differently to chemotherapy because of individual differences and ineffective anti-cancer therapy can result in the development of resistant clones [65]. Therefore, it is critical to determine the most appropriate chemotherapy individually for each patient for a successful treatment.

Studies performed with animal xenograft models showed that only a subset of cancer cells within each tumor is capable of initiating tumor growth. This subpopulation has been named as “Cancer Stem Cells (CSCs)” or “Cancer-Stem-Like Cells (CSLCs)” and the presence of CSCs has been shown in various types of cancer [66,67,68,69]. CSCs constitute less than 1% of tumor mass and maintain stem-like characteristics in that they proliferate very slowly, have a self-renewal and differentiation capacity and generate progenitors through asymmetric divisions [70]. Unlike the bulk of tumor cells, CSCs are resistant to chemo-/radio-therapy and they are mainly responsible for tumor progression, relapse and metastasis [69,71,72]. Therefore, it has been suggested that a tumor chemosensitivity/chemoresistance assay that can analyze both bulk of tumor cells and CSCs may be a very useful prognostic tool for optimal treatment selection [73].

Recently, a new drug response assay has been developed, which is called ChemoID^®^ that tests separately CSCs and bulk tumor cells. This test has been used to predict the most active/appropriate combination of chemotherapy agents to treat cancer patients individually [65,74,75,76]. Mathis et al. used the ChemoID assay to measure the sensitivity and resistance of CSCs and bulk of tumor cells cultured from two biopsies of human ependymoma challenged with several chemotherapy agents. Their results demonstrated that patients with the same histological stage and grade of cancer may vary considerably in their clinical response and ChemoID testing this could lead to more effective and personalized anti-cancer treatments in the future [65]. In the other study, Claudio et al. reported that ChemoID might be a suitable method in the determination of optimal treatment for malignancies of the central nervous system [77]. In another study conducted on head and neck cancer, three biopsy samples, which were taken from the oral cancer patients at different stages, were used in the ChemoID platform to allow growing both CSCs and bulk tumor cells. As a result, they tested different chemotherapeutics on tissue samples and found each tissue displayed different responses to drugs (no response, intermediate and sensitive). The authors suggested that chemosensitivity assays may guide clinicians to choose the best treatment options for patients, considering the individual differences and the presence of CSCs [74].

As another method that allows growing cancer stem cells, Raghavan et al. developed a 3D hanging drop spheroid platform to propagate primary cancer stem cells from ovarian cancer patients [78]. The results showed that the spheroids, which are formed by ovarian CSCs, were found to overexpress ALDH (a CSC marker), display cisplatin resistance and were successfully able to represent individual patient tumors [78].

Besides, CSCs play a major role in determining the patient prognosis not only in solid tumors but also in hematological tumors, such as leukemias. A study indicated that the higher stemness-related gene expression is positively correlated with the poorer prognosis in patients with acute myeloid leukemia [79]. Consequently, it is quite clear that the stemness (the magnitude of CSC) in individual patients should be taken into consideration for a better care of patients.

## 4. The Role of Multigene-Based and Pharmacogenetics Studies in Drug Response

Improvement in chemotherapeutic responses could be achieved by gaining more detailed information on the molecular determinants (i.e., DNA, RNA or protein) underlying this heterogeneity. Pharmacogenetics approaches can be used to integrate information on drug responsiveness with alterations in molecular entities, often on a genome-wide scale. By using information gleaned from pharmacogenetics studies, it is anticipated that cancer chemotherapy can be tailored to the individual patient or tumor phenotype [80]. Tanaka et al. developed a prediction model for the in vitro activity of eight anticancer drugs (5-Fluorouracil (5-FU), Cisplatin (CDDP), Mitomycin C (MMC), Doxorubicin (DOX), Irinotecan (CPT-11), 7-ethyl-10-hydroxycamptothecin (SN-38), Paclitaxel (TXL),and Docetaxel (TXT)), along with individual clinical responses to 5-FU using expression data of 12 genes. However, the results showed that none of the 12 selected genes alone could predict such clinical responses [81]. Matsuyama and colleagues identified some genes related to 5-fluorouracil (5-FU) sensitivity in colorectal cancer and utilized these genes for predicting the 5-FU sensitivity of liver metastases. At the end of study, four genes (*TNFRSF1B*, *SLC35F5*, *NAG-1* and *OPRT*) were found to have significantly different expression profiles in 5-FU-nonresponding and -responding tumors (*p* < 0.05). A “Response Index” system using three genes (*TNFRSF1B*, *SLC35F5* and *OPRT*) was then developed using a discriminate analysis; the results were found to be well-correlated with the individual chemosensitivities. Based on the results of Response Index system, consisting of *TNFRSF1B*, *SLC35F5* and *OPRT*, it has great potential for predicting the efficacy of 5-FU-based chemotherapy against liver metastases from colorectal cancer [82].

In another study, a chemosensitivity prediction model has been developed to assess the response to eight different chemotherapeutics (5-FU, CDDP, MMC, DOX, CPT-11, SN-38, TXL, and TXT) using 19 different cancer cell lines. Based on the prediction model, they selected five marker genes as drug sensitivity determinants and identified nine novel predictive genes to refer to sensitivity for four chemotherapeutics (5-FU, CDDP, DOX, and CPT-11 (SN-38)) [83]. These genes may provide an advantage to predict response to therapy.

In the other study conducted on breast cancer, the gene expression profile was analyzed in tumor samples of 84 patients to predict the response to combination therapy (P-FEC; paclitaxel, fluorouracil, epirubicin and cyclophosphamide). As a result, the 70-gene classifier was found to have high sensitivity and high negative predictive value in predicting pathologic complete response to P-FEC combination treatment [84]. Besides, the classifier can also use to predict the prognosis of patients with lymph-node negative and estrogen receptor-positive tumors [84].

In 2015, Xiao et al. performed a study to identify novel gene mutations related with drug metabolism in osteosarcoma and to guide individualized chemotherapy for osteosarcoma based on the analysis of expression and mutations of the genes. It has been suggested that high-throughput genotyping allows mapping of osteosarcoma mutations, and novel gene mutations represent new targets for both diagnosis and therapeutic approaches [85].

Cytotoxic activity of a drug is not a sole determinant of drug effect in a patient. The mutations in patients are also of importance. Pharmacogenetics provides inherited genetic differences in pathways that metabolize the drugs. This genetic variety is responsible for the individual responses to drugs. Some patient may metabolize the drugs faster or slower than the other patients, causing interindividual variety in responses to the drugs. Pharmacogenetics examine germline mutations (SNIPs; single gene mutations) on the enzymes that are responsible for the distribution of drugs (pharmacokinetics). Pharmacogenetics could particularly be important for the side effects/adverse reactions (ADRs) of the drugs. To know/predict the possible side effects of a drug in an individual patient is tremendously important. This is because some patients may respond to the treatment very well with minimum side effects, but the others may respond to it with a minimum benefit with a deadly side effect. Therefore, it should be critical to predict the effectiveness and possible side effects of chemotherapy before applying it. Therefore, the combination of pharmacogenetics and TCAs seems to be a wise idea.

Moreover, pharmacogenetics is also required for the precision medicine that relies on the identification of biomarkers [86,87]. Hammoudeh et al. performed next generation sequencing (NGS) using a cancer panel (Illumina) to screen for pharmacogenetic susceptibility variants in blood samples of 10 patients with papillary thyroid cancer (PTC). The results showed the TruSight Cancer Sequencing Panel to be a useful clinical tool for determination of oncotherapy-associated pharmacogenetic variants in the blood of patients [88]. Pharmacogenetics also yields information of the influence of chemotherapeutic drug-related gene polymorphisms on toxicity and survival of cancer patients [89]. This kind of information will be of immense importance in the future of personalized oncology for better outcomes. In a recent study, the relationship between genetic polymorphisms and outcomes (overall response rate, overall survival and progression-free survival) in patients treated with platinum-based chemotherapy has been investigated. It was reported that eleven polymorphisms in nine genes, including *ERCC1*, *XPA*, *XPD*, *XPG, XRCC1*, *XRCC3*, *GSTP1*, *MTHFR* and *MDR1* were found to be significantly associated with the treatment outcomes [90]. In another pharmacogenetic study, it was reported that *ABCC5* polymorphisms might explain partially the interpatient variability in doxorubicin disposition [91]. Taken together, pharmacogenetic analysis with TCAs seems to give a bigger frame of drug effect in terms of personalized oncology.

In addition to pharmacogenomics, other omics techniques have been used to predict chemotherapy response in cancer patients. Since cancer is a complex and heterogeneous disease, various abnormalities occur in the levels of DNA, RNA, protein and even metabolites. Therefore, different omics techniques such as metabolomics, proteomics, transcriptomics, and genomics and their use in cancer research have been gaining attention in recent years [92,93].

## 5. The Role of Organoid Models and 3D Culture Systems in Chemosensitivity Assays

In 1951 and 1980, a 3D histological culture of a tumor was developed from mouse breast cancer [94,95] and it was observed that these 3D culture systems preserved cell–cell interactions [95]. This model resulted in the development of histoculture drug response assays (HDRAs) for further studies [96]. In recent years, spheroids are, however, getting trendy for their use in drug development and personalized medicine.

Since the cells are grown in a 3D environment, these models can mimic tissue structure and organization. Culture of pure primary cancer cells is expected to provide an innovative platform for developing personalized medicine. The cancer tissue-originated spheroids can be cultured and expanded ex vivo using 3D culture systems and special culture medium to propagate cancer stem cells. This method is thought to evaluate the chemosensitivity in cancer cells obtained from cancer cases.

Weiswald et al. studied patient-derived colorectal cancer xenografts that produce colospheres. They reported that the colospheres were correlated with tumor aggressiveness [96]. They observed colospheres closely mimic biological characteristics of in vivo colorectal tumors. Primary human tumor culture models allow person-specific drug susceptibility tests, and thereby they allow one to employ personalized treatments for cancer patients. In our lab, we have been performing 3D culture tumor models by using tumor samples and culturing cells in ultralow conditions and special culture medium. Figure 4 shows some representative pictures from our studies.

In vitro tumor organoid models could potentially be used to evaluate the sensitivity to various classes of chemotherapeutics [97]. An organoid model obtained from patient tumor tissue could be exposed to an empirically chosen therapeutic, and the results could be used to predict tumor response. A recent study showed that patient-derived organoid models can successfully predict sensitivity to chemotherapy and radiotherapy in patient with metastatic colorectal cancer [98]. An extensive review, which includes 60 studies in organoid research, indicated that organoid cultures are predictors of chemotherapy response in patients with different types of cancer [99].

Organoids are even thought to mimic metastasis. The genetic analysis showed that organoids reflect the metastasis from which they were derived. Furthermore, ninety percent of somatic mutations were shared between organoids and biopsies from the same patient, and the DNA copy number profiles of organoids and the corresponding original tumor show a correlation of 0.89, implying patient-derived organoids are an ex vivo platform to personalize anti-cancer treatment [100]. 3D cultures are clearly superior to 2D cultures. Normal and malignant epithelial cells appear phenotypically similar under 2D culture conditions. However, they behave differently when grown as 3D cultures [101]. Besides, 3D tumor models have an advantage in that they mimic the tumor microenvironment by including the components of the tumor microenvironment such as fibroblasts and immune cells [102,103].

High-content screening (HCS) is frequently used at the pre-clinical stage of drug discovery due to its relatively low cost and larger and faster data output compared with animal studies [104]. However, 2D cultures are thought to display increased drug sensitivity compared with corresponding 3D tumor spheroids, which may be a disadvantage for HCSs [105].

Patient-derived xenograft (PDX) models are also good candidates for the personalized chemotherapy. They preserve primary tumor characteristics including tumor histology, vascularization, and structure [106]. Above all, they can also mirror the heterogeneity of tumors. In PDX models, the tumor microenvironment is available and contains both cancer cells and various stromal cells, such as fibroblasts and mesenchymal stem cells as well as endothelial cells and macrophages [102]. All these non-malignant cells are thought to support tumor growth via by various cytokines (e.g., IL-6). A vicious interaction has been known to exist between tumor cells and their microenvironment, which supports the growth of each cell type and closely mimics tumor structure.

## 6. The Importance Circulating Tumor Cells in Chemosensitivity Assays

Circulating tumor cells (CTCs) are a subpopulation of cancer cells, and they originate from primary or metastatic tumors and can travel in the bloodstream. CTCs are obtained from blood (via a liquid biopsy) and can be used for different analyses [107].

Similar to solid tumor samples, CTCs can be used to predict response to treatment. In a study conducted on 106 patients with unresectable colorectal liver metastasis, it was reported that a multidisciplinary treatment option that involves hepatic artery infusion with chemo-filtration, and chemotherapeutics selected by liquid biopsy precision oncotherapy is a safe and efficacious alternative for patients [108]. Similarly, in three other studies, CTCs were subjected to chemosensitivity assays to predict response to chemotherapeutics in patients with recurrent cutaneous melanoma (with locoregional pelvic metastases) [109], patients with unresectable recurrent rectal cancer [110], and patients with recurrent ovarian cancer [111].

Since the sample needed for analysis is taken from blood, liquid biopsy is accepted as less invasive than traditional biopsy techniques. On the other hand, these cells rarely exist in blood circulation—as few as one CTC per billion hematological cells, thus detection of CTCs is technically challenging [112]. Fortunately, recent developments have been focusing on increasing the number of captured CTCs and it may allow widespread use of this technology in the near future.

## 7. TCA and Its Relation with Response to Treatment and Clinical Outcomes

TCAs have acceptable correlation levels with response to treatment. It was reported that, based on cut-off values determined empirically, the test accurately predicted resistance for 36 of 41 patients (88%) who did not respond to the drug [113]. It also predicted sensitivity with a high degree of accuracy—21 of 22 patients (95%) who responded to the drug tested had a sensitive assay [113]. Therefore, TCAs accurately predict both sensitivity and resistance. In another study, executed by Konecny et al., on primary epithelial ovarian cancer with FIGO stage III, it was stated that they found a correlation with the test results and clinical response [15]. They also found a correlation with progression-free and overall survival [15]. Mehta et al. performed a study to determine if TCA (in vitro extreme drug resistance (EDR) assay in this study) results for 103 breast cancer patients were related with clinical outcome following chemotherapy [114]. They stated that the EDR scores were significantly associated with time to tumor progression and overall survival. There is another study in which EDR was performed as a TCA. They determined platinum resistance in ovarian cancer cases (stages IIC, III, and IV) and compared their clinical characteristics [55]. They found that patients with tumors showing EDR to platinum were at significantly increased risk for progression and death when treated with standard platinum-based regimens. For TCAs, resistance data is probably more reliable than sensitivity data. This is more likely to be explained by the different in vitro and in vivo conditions. In the case of finding a sensitive drug, TCAs do not mimic the whole organism that plays a role in the drug action by employing liver pass, renal clearance, immune system effect, etc. All these factors should make an impact on the drug activity by diminishing the effect of drug. Therefore, the sensitivity may be overestimated (false increase in drug activity) in the case of TCAs. However, when it comes to finding the resistant drug, the cell in TCAs is directly exposed to the drugs, without taking advantage of other activity-diminishing factors. Therefore, if a cell does not die under TCA conditions, it is highly possible to expect it being excessively resistant in the body. There are more studies showing the good correlation between EDR and clinical outcomes (e.g., progression-free survival) [15,55,115]. Patients with extreme EDR to platinum compounds in vitro had a lower survival time compared with those with low to intermediate drug resistance [55].

Numerous clinical trials examining the relationship between in vitro tumor response and clinical outcomes are available. These data suggest that in vitro drug response assays can accurately predict drug resistance and can identify patients who are more or less likely to benefit from a given agent [116]. These highly promising results make it possible to design tailored (patient-specific) regimens for each different patients. By eliminating ineffective agents, the patient is spared toxic treatment without benefit, and the selection of agents active in vitro may increase the probability of response [116]. Moreover, Whitehouse et al. [6] performed ATP-TCA on colorectal cancer tissues with various drugs (5-FU, irinotecan, oxaliplatin by itself) and drug combinations (oxaliplatin + 5-FU). The drugs and combinations found effective in the assay are similar to those found to be active in clinical trials, suggesting that TCA may actually be able to predict sensitivity and resistance to chemotherapy in individual patients [6]. It is also possible to predict the gemcitabine sensitivity by employing a TCA. In a study in pancreatic cancer, it was found that gemcitabine sensitivity measured by TCA was well-correlated with in vivo drug activity [12]. Biomarkers are also used to predict the effectiveness of anti-cancer drugs. However, in a comparative study, it was reported that TCA was found to have superiority to the biomarkers in terms of the prediction of chemosensitivity to drugs used [117]. Moreover, bioengineered 3D cancer models are reported to be a reliable preclinical patient-specific platform as drug testing assays [118].

## 8. Concluding Remarks

Personalization of chemotherapy in cancer is inevitable due to the vast majority of newly-approved chemotherapeutic drugs (e.g., small molecules) in recent years. However, oncologists eagerly demand to know which drug would be suitable for which patient. Therefore, TCAs are supposed to gain more importance to provide oncologists with data to be used for this specific task. Moreover, totally new TCA-based technologies (e.g., microfluidics) are expected to be developed quickly.

## Figures and Tables

**Figure 1 medicina-57-00636-f001:**
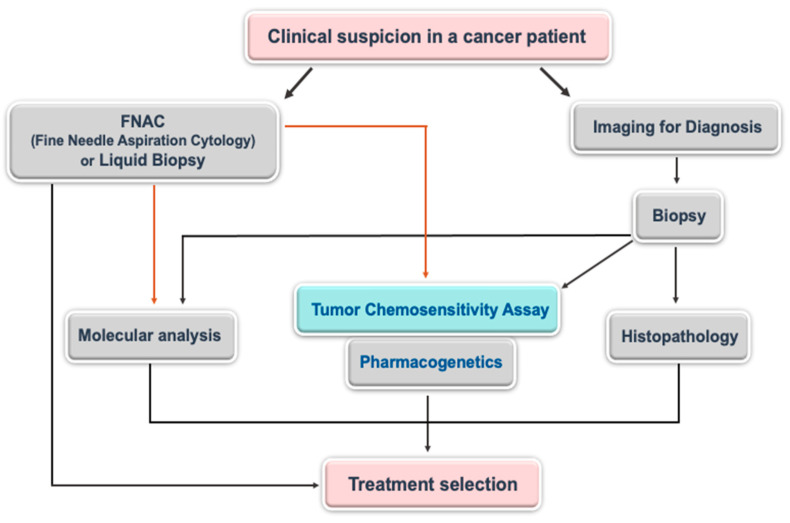
The possible place of tumor chemosensitivity assay (TCA) and pharmacogenetics in the treatment of cancer patients.

**Figure 2 medicina-57-00636-f002:**
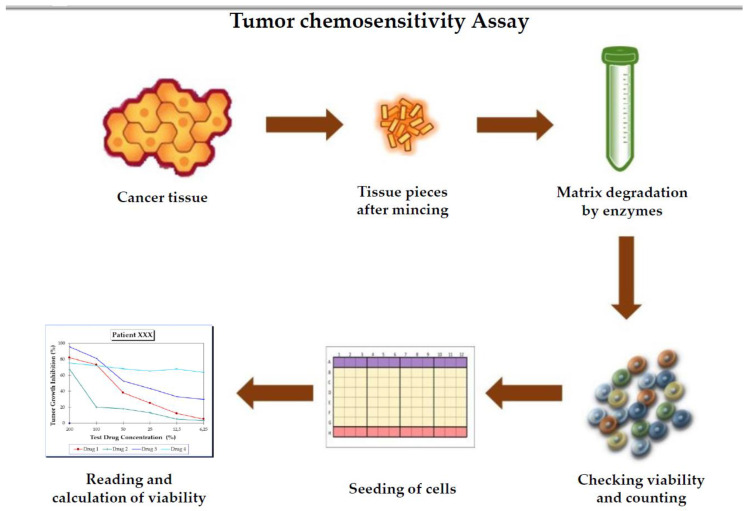
The steps of a TCA, starting from the obtaining of tissue to the assessment of cytotoxicity.

**Figure 3 medicina-57-00636-f003:**
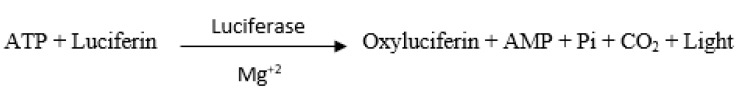
Reduction of luciferin to oxyluciferin via luciferase in the presence of ATP and Mg^2+^ and formation of bioluminescent light that is correlated with the amount of intracellular ATP.

**Figure 4 medicina-57-00636-f004:**
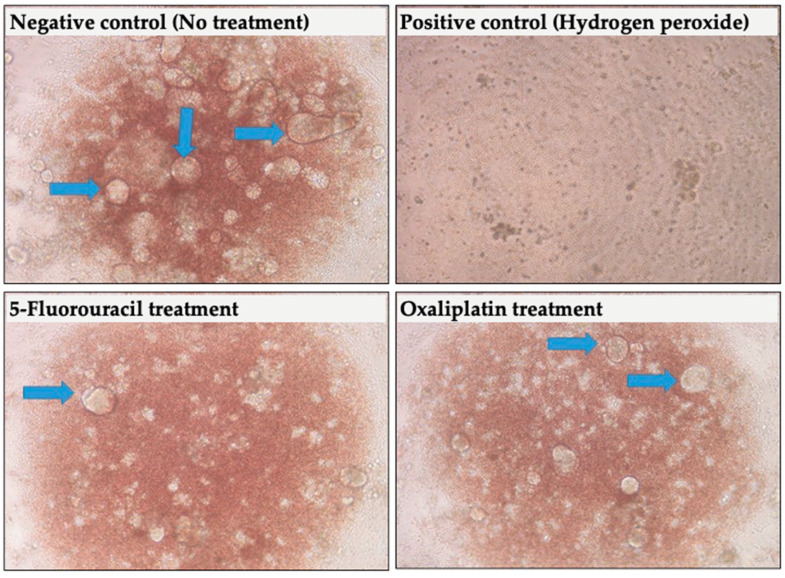
Phase contrast microscopy images of sphere-like structures and their responses to chemotherapeutics in a TCA platform. The treatments resulted in varying degrees of toxic effect that is evident by the less and varying number of sphere-like structures. In the positive control (by hydrogen peroxide -H_2_O_2_-), there is no sphere-like structure.

**Table 1 medicina-57-00636-t001:** The tumor chemosensitivity assays.

TCA Methods	References
Human Tumor Clonogenic Assay (HTCA)	[18,19,20,21]
3-(4,5-Dimethylthiazol-2-yl)-2,5-Diphenyltetrazolium Bromide (MTT) Assay	[16,28,29,30,31,32]
Adenosine Triphosphate (ATP) Assay	[41,42,43,44,45,46,47]
Extreme Drug Resistance Assay	[53,54,55]
Tissue Explant Assay (Histodrug Response Assay, HDRA)	[56,57]
Differential Staining Cytotoxicity Assay (DISC)	[26,58,59]
Fluorescent Cytoprint Assay (FCA)	[60,61]
Collagen Gel Droplet-Embedded Culture Drug Sensitivity Test (CD-DST)	[62,63,64]

## Data Availability

Not applicable.

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
