# Peer review of "Tumor Chemosensitivity Assays Are Helpful for Personalized Cytotoxic Treatments in Cancer Patients"

_medicina, 2021, doi:10.3390/medicina57060636_

Round 1

Reviewer 1 Report

The authors provide a very comprehensive overview of the different Tumor Chemosensitivity Assays in use over the years, whilst emphasizing on their relevance in identifying patients that will benefit from a specific chemotherapeutic agent.

Specific Major comments: None

Specific Minor comments:

  • The authors are suggested to include a section in their Introduction where it is clear for the reader that The objective of this review, is to ... And also add a section at the end of the review entitled Concluding remarks

Non-specialist readers and clinicians are mostly interested in the broader picture, so structuring the review such that it has clearly stated objectives and conclusions helps in terms of clarity.

  • The authors do not only provide an overview of the literature, but also present their own opinions as to where they think current efforts need to be focused. I refer specifically to their views on combining TCA assays with OMICs technologies. This is something that the authors could emphasize more in their concluding remarks ... for example the potential of TCA assays based on 3D organoid models/OMICs screening

Author Response

The responses have been attached as a word file.

Reviewer 2 Report

This review is very interesting and important not only for Researches but also for Clinicians involved in Precision Oncotherapy.

However, the present version of the manuscript requires, in my opinion, a Major Revision.

Major changes suggested

  1. Include a chapter regarding liquid biopsies, circulating tumor cells (CTCs), in vitro cultures for chemosensitivity tests.
  2. Figure 1 should include chemosensitivity tests (and pharmacogenetics tests) on circulating tumor cells (CTCs) isolated using liquid biopsies.
  3. Elaborate implications of CTCs chemosensitivity tests on tailored chemotherapy.
  4. Chapter 6. TCA and Its Relation with Response to Treatment and Clinical Outcomes, is very important but references reported (107, 15, 108, 109, 110, etc.) are too old. This problem has implications on the concepts elaborated.
  5. More recent references are necessary for such an interesting review.

Minor changes suggested

  1. Regarding chemosensitivity test in other types of tumor cells, please onsider references as:

Real-life multidisciplinary treatment for unresectable colorectal cancer liver metastases including hepatic artery infusion with chemo-filtration and liquid biopsy precision oncotherapy. Observational cohort study. Guadagni S, Clementi M, Mackay AR, Ricevuto E, Fiorentini G, Sarti D, Palumbo P, Apostolou P, Papasotiriou I, Masedu F, Valenti M, Giordano AV, Bruera G. Journal of Cancer Research and Clinical Oncology 2020; 146:1273–1290. DOI: 10.1007/s00432-020-03156-3

A pilot study of the predictive potential of chemosensitivity and gene expression assays using circulating tumour cells from patients with recurrent ovarian cancer.Guadagni S, Clementi M, Masedu F, Fiorentini G, Sarti D, Deraco M, Kusamura S, Papasotiriou I, Apostolou P, Aigner KR, Zavattieri G, Farina AR, Vizzielli G, Scambia G, Mackay AR. International Journal of Molecular Sciences 2020; 21, 4813; doi:10.3390/ijms21134813

Circulating tumour cell liquid biopsy in selecting therapy for recurrent cutaneous melanoma with locoregional pelvic metastases: a pilot study. Guadagni, S., Fiorentini, G., Papasotiriou, I, Apostolou P, Masedu F, Sarti D, Farina AR, Mackay AR, Clementi M. BMC Res Notes 2020;13: 176. https://doi.org/10.1186/s13104-020-05021-5

Precision oncotherapy based on liquid biopsies in multidisciplinary treatment of unresectable recurrent rectal cancer: a retrospective cohort study. Guadagni S, Fiorentini G, De Simone M, Masedu F, Zoras O, Mackay AR, Sarti D, Papasotiriou I, Apostolou P, Catarci M, ClementiM, Ricevuto E, Bruera G. Journal of Cancer Research and Clinical Oncology 2020; 146:205-219.

  1. Improve the editing: i.e. page 11, line 254
  2. English grammar: i.e page 2, line 41…liquid
  3. Page 2, lines 47-48: too old references concerning negative and positive predictive values of chemosensitive tests on tumor cells.

Author Response

The responses have been added as a word file.

Round 2

Reviewer 2 Report

Good revision. Congratulations.